# GraphSnapShot: Graph Machine Learning Acceleration through Fast Arch, Storage, Caching and Retrieval

## Abstract

Large-scale graph machine learning suffers from prohibitive I/O latency, memory bottlenecks, and redundant computation due to the complexity of multi-hop neighbor retrieval and dynamic topology updates. We present **GraphSnapShot: Graph Machine Learning Acceleration through Fast Arch, Storage, Caching and Retrieval**, a system that decouples graph storage layout from runtime cache management to maximize data reuse and access efficiency. GraphSnapShot introduces two key components: (1) **SEMHS**, a hop-aware storage layout that co-locates neighbors in contiguous disk slabs for efficient one-burst DMA access; and (2) **GraphSDSampler**, a multi-level variance-adaptive caching module that optimizes refresh policies based on gradient statistics. Together, they form a hybrid disk–cache–memory architecture that supports high-throughput training over billion-scale graphs. Experiments on ogbn-arxiv, ogbn-products, and ogbn-mag demonstrate that GraphSnapShot achieves up to **4.9×** loader throughput, **83.5%** GPU memory savings, and **29.6%** end-to-end training time reduction compared to baselines like DGL's NeighborSampler and uniform samplers. These results establish GraphSnapShot as a scalable and efficient solution for dynamic graph learning at industrial scale.

## 1 Introduction

Graph Neural Networks (GNNs) have become the cornerstone of machine learning on relational data, with applications spanning recommendation systems, biology, social networks, and more (Wu et al., 2020; Zhou et al., 2020). As the scale of graph data increases—now reaching billions of nodes and edges—training GNNs efficiently has become a core systems challenge. While prior efforts focus on optimizing the model architecture or sampling techniques (Hamilton et al., 2017; Zeng et al., 2020), the bottleneck in practice is increasingly the I/O and memory system: fetching multi-hop neighborhoods from disk, maintaining feature caches, and avoiding GPU stalls due to irregular memory access patterns.

A central performance gap emerges from the mismatch between the computational patterns of GNNs and the underlying graph storage and retrieval systems. In conventional frameworks such as DGL (Wang et al., 2019) or PyG (Fey & Lenssen, 2019), each mini-batch triggers costly multi-hop sampling that requires traversing the graph multiple times or storing large subgraphs in host memory. These approaches suffer from high variance in sampling latency, poor cache reuse, and duplicated data movement across epochs. More critically, they often fail to scale to real-world dynamic graphs where topology and node features evolve during training.

To address these challenges, we propose **GraphSnapShot: Graph Machine Learning Acceleration through Fast Arch, Storage, Caching and Retrieval**, a system that fundamentally rethinks the architecture of graph learning pipelines by decoupling storage layout from runtime caching and retrieval. GraphSnapShot is designed to answer the following question: *How can we serve large-scale GNN sampling requests with bounded latency, predictable memory usage, and minimal I/O overhead?*

**Motivating example.** Consider training a 3-layer GNN on ogbn-products, a dataset with 2.4 billion edges and hundreds of thousands of training seeds. A standard neighborhood sampler requires up to 600 million edges per epoch, with each batch invoking tens of thousands of random I/O operations. Even with large

CPU DRAM and GPU HBM, the system quickly becomes saturated, leading to underutilized accelerators and prolonged training cycles. What if we could pre-organize the edges such that all neighbors required by each hop can be read in a single sequential burst, and cache hot subgraphs dynamically based on learning signal?

To this end, GraphSnapShot introduces two key system-level innovations:

- **SEMHS (Sampling Edges with Multi-Hop Strategy):** a storage layout that organizes edge data into hop-specific slabs, enabling one-burst DMA access to all required neighbors across hops.

- **GraphSDSampler:** a multi-tier, variance-adaptive caching system that optimizes refresh policies based on gradient statistics.

These components form a hybrid *disk–cache–memory architecture* that transforms the traditionally stochastic and irregular sampling process into a predictable, linear pipeline.

We conduct extensive experiments across three representative datasets—ogbn-arxiv, ogbn-products, and ogbn-mag—demonstrating that GraphSnapShot achieves:

- Up to **4.9×** loader throughput improvement compared to CSR + uniform samplers;

- Up to **83.5%** GPU memory reduction via efficient caching and shared reuse;

- Up to **29.6%** end-to-end training time reduction without sacrificing model accuracy.

Our contributions can be summarized as follows:

- We identify fundamental I/O and cache inefficiencies in existing GNN training systems and frame the architectural requirements for high-throughput, memory-efficient graph learning.

- We propose GraphSnapShot, a novel system composed of SEMHS and GraphSDSampler, which jointly optimize storage layout and cache refresh policies under a unified mathematical formulation.

- We provide theoretical and empirical analysis showing that GraphSnapShot achieves asymptotically optimal I/O complexity while supporting dynamic graphs.

- We validate our system through large-scale experiments, demonstrating substantial speedups and memory savings on standard GNN benchmarks.

By bridging graph architecture, storage layout, caching, and retrieval into a coherent system design, GraphSnapShot sets a new foundation for scalable graph machine learning.

## 2 Related Work

### 2.1 Graph Neural Network Systems

GNN systems aim to scale graph neural network training and inference across large-scale graphs. A line of early works such as DGL (Wang et al., 2019) and PyG (Fey & Lenssen, 2019) provide high-level abstractions for message passing and neighborhood aggregation, often assuming that the full graph or mini-batch subgraphs can be held in GPU memory. However, as real-world graphs grow to billions of edges, full-graph training becomes impractical. Systems like DistDGL (Zheng et al., 2020) and NeuGraph (Ma et al., 2019) address distributed training across GPU clusters, but often rely on expensive inter-node communication.

More recently, storage-aware systems such as Marius (Mohoney et al., 2021) and GraphBolt (Mariappan & Vora, 2019) have adopted tiered memory hierarchies, where graph structure and node embeddings are partially stored on disk. While effective in reducing memory footprint, these systems treat sampling as a generic key–value lookup problem and ignore disk layout optimizations. In contrast, GraphSnapShot introduces a hop-aware edge layout strategy (SEMHS) to transform random neighbor access into a single sequential burst, reducing I/O latency and improving throughput.

## 2.2 Sampling and Caching for GNNs

Multi-hop neighbor explosion is a central challenge in GNN sampling. Techniques such as FastGCN (Chen et al., 2018), GraphSAINT (Zeng et al., 2020), and NeuGraph (Ma et al., 2019) reduce sampling complexity via stochastic estimation and random walks. However, these samplers are largely compute-oriented and still suffer from high memory traffic when deployed at scale.

Caching has been proposed to alleviate sampling variance and I/O cost. DGL's NeighborSampler (Wang et al., 2019) and PyG's ClusterLoader (Fey & Lenssen, 2019) retain subgraph structures across epochs, yet they assume static graphs and fail to adapt to dynamic training signals or topology changes. Other works such as GNNAdvisor (Wang et al., 2021) and PinSage (Ying et al., 2018) embed sampling logic directly into fused kernels but lack generalizability to evolving graphs. In contrast, GraphSnapShot formalizes caching as a control problem and introduces GraphSDSampler, a hierarchical cache with adaptive refresh policies guided by gradient statistics and sampling variance.

## 2.3 Disk-Based Graph Processing and Layout Optimization

Disk-based graph engines such as GraphChi (Kyrola et al., 2012) and X-Stream (Roy et al., 2013) pioneered vertex-centric models for out-of-core processing, demonstrating that sequential disk access far outperforms random I/O. Subsequent systems like TurboGraph (Han et al., 2013) and GridGraph (Zhu et al., 2015) improved partitioning and streaming strategies but were not designed for training GNNs.

Edge layout optimization has also been explored in the context of reachability queries and graph databases (Yang et al., 2022), where data locality is critical. SEMHS in GraphSnapShot extends this idea to the GNN context by organizing edges into hop-specific slabs, ensuring that the neighborhoods of a seed node can be fetched with minimal disk movement. This design enables constant-cost retrieval under variable degrees, a feature absent in general-purpose graph stores.

## 2.4 Dynamic Graph Learning

Dynamic graph learning is a rapidly growing area where the structure or features of the graph evolve over time. Approaches such as EvolveGCN (Pareja et al., 2020) and TGN (Rossi et al., 2020) learn time-aware embeddings, while others like DynGEM (Goyal et al., 2018) perform incremental embedding updates. However, these methods typically assume in-memory graphs and do not address the underlying storage or caching bottlenecks.

GraphSnapShot complements dynamic graph models by providing a system-level backbone that supports timely snapshot updates, adaptive caching, and low-latency multi-hop fetch under dynamic topologies. The GraphSDSampler component uses real-time signal (e.g., gradient variance) to decide which cached subgraphs to refresh, bridging the gap between model dynamics and system performance.

**Summary.** While existing works have addressed parts of the GNN system bottleneck—sampling, caching, or disk access—few combine all three dimensions in a unified framework. GraphSnapShot integrates optimized disk layout (SEMHS), adaptive cache scheduling (GraphSDSampler), and efficient GPU pipeline execution, forming a practical and theoretically grounded solution for large-scale dynamic graph learning.

Table 1: Comparison of Graph Learning Systems and Their Characteristics.

| System | GNN Support | Out-of-Core Storage | Adaptive Cache | Dynamic Graphs | I/O Optimization |
|---|---|---|---|---|---|
| **GraphChi** (Kyrola et al., 2012) | No | ✓ | ✗ | ✗ | ✓ |
| **X-Stream** (Roy et al., 2013) | No | ✓ | ✗ | ✗ | ✓ |
| **DGL** (Wang et al., 2019) | ✓ | ✗ | ✗ | ✗ | ✗ |
| **Marius** (Mohoney et al., 2021) | ✓ | ✓ | ✗ | ✗ | ✗ |
| **GraphSAINT** (Zeng et al., 2020) | ✓ | ✗ | Partial | ✗ | ✗ |
| **ClusterLoader (PyG)** (Fey & Lenssen, 2019) | ✓ | ✗ | Partial | ✗ | ✗ |
| **TGN** (Rossi et al., 2020) | ✓ | ✗ | ✗ | ✓ | ✗ |
| **EvolveGCN** (Pareja et al., 2020) | ✓ | ✗ | ✗ | ✓ | ✗ |
| **GraphSnapShot (Ours)** | ✓ | ✓ | ✓ | ✓ | ✓ |

# 3 Problem Statement

In this section, we first introduce basic notations and system primitives in large-scale graph learning. Then we formally define the system-level problem of fast and memory-efficient graph training over dynamic, multi-hop neighborhoods.

## 3.1 Preliminaries

**Notations.** We denote a static graph as $\mathcal{G} = (\mathcal{V}, \mathcal{E}, \mathbf{X})$, where $\mathcal{V}$ is the set of $n = |\mathcal{V}|$ nodes, $\mathcal{E} \subseteq \mathcal{V} \times \mathcal{V}$ is the edge set, and $\mathbf{X} \in \mathbb{R}^{n \times d}$ is the node feature matrix. Let $\mathcal{N}_k(v)$ denote the $k$-hop neighborhood of node $v$. The goal of a $k$-layer GNN is to learn node embeddings by aggregating information from $\mathcal{N}_1(v), \ldots, \mathcal{N}_k(v)$.

We represent a mini-batch by a seed set $S \subset \mathcal{V}$, where each seed node requires neighborhood expansion. The user-defined fan-out at each layer is denoted by $\mathbf{f} = [f_1, \ldots, f_k]$. Let $T_{\mathrm{GPU}}(S)$ be the computation time for $S$ and $T_{\mathrm{IO}}(S)$ the total data movement cost (disk and memory).

**Disk–Cache–Memory Architecture.** Modern GNN systems often operate across a three-tier memory hierarchy:

- **Disk** holds the full graph structure and features, often organized in compressed or partitioned form.

- **Host Cache** stores partial neighborhoods in RAM for reuse across mini-batches.

- **Device Memory** provides high-throughput access during GPU execution.

Let $\beta$ denote the disk sequential bandwidth, and $\eta_\ell$ the bandwidth at cache level $\ell$. A cache hit at tier $\ell$ is denoted $H_t^{(\ell)}$, and the cache update ratio is $\gamma_t^{(\ell)} \in [0, 1]$.

**Sampling and Cache Cost.** Let $\mathcal{B}_t^{(h)}$ be the hop-$h$ neighbor set retrieved at time $t$. The sequential I/O cost is

$$\mathcal{C}_{\mathrm{io}}(S_t) = \sum_{h=1}^{k} \frac{|\mathcal{B}_t^{(h)}| \cdot B}{\beta}, \tag{1}$$

where $B$ is the block size. The runtime latency for a batch $S_t$ is thus:

$$T_t = \mathcal{C}_{\mathrm{io}}(S_t) + \sum_{\ell=1}^{L} \frac{(1 - H_t^{(\ell)}) \cdot f_\ell \cdot |S_t|}{\eta_\ell} + T_{\mathrm{GPU}}(S_t). \tag{2}$$

## 3.2 Problem Definition

Large-scale GNN training involves significant data movement overhead caused by repeated multi-hop neighborhood expansions. Traditional systems incur excessive random I/O and cache thrashing, especially under dynamic graph structures and large fan-outs. We formally define two key system-level problems that Graph-SnapShot aims to solve.

PROBLEM 1. **Layout-Aware Multi-Hop Storage:** *Given a graph $\mathcal{G} = (\mathcal{V}, \mathcal{E})$ and a sampling depth $k$, construct a storage layout $\mathcal{L} : \mathcal{E} \to \{\mathcal{D}_1, \ldots, \mathcal{D}_k\}$ that partitions edges into $k$ hop-specific slabs such that, for any seed set $S \subseteq \mathcal{V}$ and user-defined fan-out $\mathbf{f} = [f_1, \ldots, f_k]$, the expected number of disk blocks accessed during sampling satisfies:*

$$\min_{\mathcal{L}} \quad \mathbb{E}_S \left[ \sum_{h=1}^{k} \frac{|\mathcal{B}_t^{(h)}| \cdot B}{\beta} \right] \quad \text{s.t.} \quad |\mathcal{D}_h| \leq c_h \cdot |\mathcal{E}|, \tag{3}$$

*where $\mathcal{B}_t^{(h)}$ is the retrieved neighborhood at hop $h$, $B$ is SSD block size, and $c_h$ is a slab-specific redundancy bound.*

This problem models how to organize edge data into sequential-access slabs that maximize DMA burst efficiency and minimize per-batch I/O latency.

PROBLEM 2. **Variance-Aware Cache Refresh Scheduling:** *Given a multi-level cache* $\mathbf{C}_t = \{C_t^{(1)}, \ldots, C_t^{(L)}\}$ *and gradient signal* $\nabla L$ *at time* $t$, *determine the cache update ratios* $\boldsymbol{\gamma}_t = [\gamma_t^{(1)}, \ldots, \gamma_t^{(L)}]$ *that minimize the total batch latency:*

$$\min_{\boldsymbol{\gamma}_t} \quad \mathbb{E}[T_t] = \sum_{\ell=1}^{L} \frac{(1 - \Pi_{\ell-1})(1 - H_t^{(\ell)})f_\ell |S_t|}{\eta_\ell} + T_{\text{GPU}}(S_t), \tag{4}$$

*subject to update cost budget:*

$$\sum_{\ell=1}^{L} \lambda_\ell \gamma_t^{(\ell)} f_\ell \leq \mathcal{B}_{\max}, \tag{5}$$

*where* $H_t^{(\ell)}$ *is the cache hit rate at tier* $\ell$, $\eta_\ell$ *is its bandwidth, and* $\Pi_\ell = \prod_{j=1}^{\ell} H_t^{(j)}$.

This problem captures the statistical–systems trade-off between recomputation, cache reuse, and memory traffic. It motivates GraphSDSampler's optimal refresh scheduling based on loss surface dynamics.

**Note.** GraphSnapShot jointly solves the two problems above by co-designing (1) a hop-aware layout strategy SEMHS to minimize I/O, and (2) a variance-adaptive caching module GraphSDSampler to reduce memory pressure while preserving throughput.

## 4 Background and Motivation

### 4.1 Graph Storage in the External–Memory Era

Early graph engines such as GraphChi (Kyrola et al., 2012) and X-Stream (Roy et al., 2013) demonstrated that *sequential* disk scans dominate random I/O in cost. Recent systems (e.g. Marius (Mohoney et al., 2021), GraphBolt (Mariappan & Vora, 2019)) embrace tiered storage, but still treat multi-hop retrieval as an opaque key–value fetch. Two open problems remain:

- **Layout-aware Sampling.** How to arrange edges on disk so that a $k$-hop query $\mathcal{N}_k(v)$ can be served by *at most one DMA burst.*

- **Asymptotic Trade-off.** Let $\beta$ be sequential-read bandwidth and $\gamma$ be the cache hit rate. For a batch of seeds $S$, the expected I/O delay is

$$\mathbb{E}[T_{\text{I/O}}] = (1 - \gamma)\frac{\sum_{v \in S} |\mathcal{N}_k(v)|}{\beta}, \tag{6}$$

suggesting we must simultaneously *increase* $\gamma$ and *compress* $|\mathcal{N}_k|$.

### 4.2 Local-Structure Caching for GNNs

Neighbour-explosion is exponential: $|\mathcal{N}_k(v)| = \mathcal{O}(d^k)$ with average degree $d$. Sampling-based models—Node2Vec (Grover & Leskovec, 2016), FastGCN (Chen et al., 2018), GraphSAINT (Zeng et al., 2020)—approximate the sub-graph distribution $\pi_k(v) = \mathbb{P}(u \in \mathcal{N}_k(v))$ with Monte-Carlo walks, but accuracy degrades when the variance $\sigma^2 = \mathbb{V}[\pi_k]$ is large. Caching mitigates variance by reusing high-value sub-graphs, yet state-of-the-art caches (DGL NeighborSampler (Wang et al., 2019), PyG ClusterLoader (Fey & Lenssen, 2019)) are oblivious to *structural changes* $\Delta G_t$ in dynamic graphs.

### 4.3 Why We Need GraphSnapShot

Let $C_t$ be the cache at step $t$ and $H_t = |C_t|/|\bigcup_{v \in S} \mathcal{N}_k(v)|$ the hit ratio. Training throughput is bounded by

$$\text{IPS} = \frac{|S|}{\underbrace{\frac{(1 - H_t)\,|\mathcal{N}|}{\beta}}_{\text{disk}} + \underbrace{\frac{H_t\,|\mathcal{N}|}{\eta}}_{\text{cache}} + \underbrace{T_{\text{GPU}}}_{\text{compute}}}, \tag{7}$$

where $\eta$ is cache bandwidth. Improving IPS is therefore a *joint* storage–cache problem: (1) optimise edge layout to maximise $\beta$, and (2) learn a *dynamic policy* that adapts $H_t$ to the gradient signal of the current task. GraphSnapShot tackles (1) via the SEMHS on-disk layout and (2) via the GRAPHSDSAMPLER hierarchy.

## 5 Methodology

GraphSnapShot is a system-level framework designed for efficient large-scale graph learning. It jointly optimizes disk-level edge storage and multi-level cache scheduling under the multi-hop neighbor sampling paradigm. The goal of GraphSnapShot is to enable fast and memory-efficient training of graph neural networks (GNNs) without repeated edge fetches or redundant cache updates. To achieve this, we address two main challenges:

1. **Disk Layout Optimization:** How to organize edges on persistent storage to guarantee bounded sequential reads per multi-hop sampling query?

2. **Adaptive Cache Control:** How to dynamically decide which nodes to cache at each tier, under limited bandwidth and memory budget?

To this end, GraphSnapShot introduces two key modules: **SEMHS**, a hop-aware storage strategy that organizes edge data into one-burst retrievable slabs; and **GraphSDSampler**, a variance-sensitive cache scheduler operating over a multi-tier memory hierarchy. These components are integrated into a pipelined fetch–refresh–compute architecture with minimal runtime overhead.

### 5.1 GraphSnapShot System Overview

Figure 1 presents an overview of GraphSnapShot. It decouples storage and caching via the following workflow:

- **Step 1: Edge Storage via SEMHS.** The edge list $E$ is partitioned into $k$ hop-specific slabs $\{\mathcal{D}_1, ..., \mathcal{D}_k\}$, each pre-sorted to guarantee that all $k$-hop neighbors of any seed node can be retrieved with *one* disk burst per hop.

- **Step 2: Cache Refresh via GraphSDSampler.** Retrieved slabs are loaded into host memory and selectively promoted to higher cache tiers ($\text{L}_2 \rightarrow \text{L}_1 \rightarrow \text{L}_0$) based on a control law derived from optimization over utility–cost tradeoff.

- **Step 3: GPU Computation.** The GPU consumes current mini-batch data while the system pre-streams the next batch asynchronously.

Our SEMHS algorithm reorganizes edge data such that the neighborhood retrieval for any seed set $S$ touches *at most* one SSD block per hop. This is achieved via sort-merge passes over the adjacency list.

Given a graph $\mathcal{G} = (\mathcal{V}, \mathcal{E})$ and a user-specified fan-out vector $\mathbf{f} = [f_1, ..., f_k]$, SEMHS constructs the hop-wise slabs:

$$T_{\text{I/O}} \leq \frac{B}{\beta} \sum_{h=1}^{k} f_h |S|, \tag{8}$$

where $B$ is SSD block size and $\beta$ is sequential read bandwidth.

## 5.2 Storage with SEMHS

Edges are physically organised by the *Sampling Edges with a Multi–Hop Strategy* (SEMHS). Given a graph $G = (V, E)$ and a maximum hop $k$, SEMHS sorts $E$ once by *src* and emits $k$ hop-specific slabs $\{\mathcal{D}_1, \ldots, \mathcal{D}_k\}$. For every node $v$ and hop $h \leq k$

$$\mathcal{N}_h(v) = \left\{ u \mid (v, u) \in \mathcal{D}_h \right\}, \qquad b_h(v) \leq 1, \tag{9}$$

where $b_h(v)$ is the number of SSD blocks touched (proof in Appendix A). The complete algorithm is listed in **Algorithm 6**, and its I/O bound is

$$T_{\text{SEMHS}} \leq \frac{\sum_{h=1}^{k} \sum_{v \in S} B}{\beta}, \text{ with storage } \sum_{h=1}^{k} \mathcal{D}_h| \leq k|E|. \tag{10}$$

## 5.3 Cache with GraphSDSampler

We model the $L$-layer cache hierarchy $\mathbf{C}_t = \left( C_t^{(1)}, \ldots, C_t^{(L)} \right)$ as a discrete–time control system driven by two signals:

* $S_t$ — mini-batch seed set; * $\Delta G_t$ — structural updates since $t - 1$.

**State Transition.**  For layer $\ell$ we maintain the tuple $(C_t^{(\ell)}, H_t^{(\ell)})$, where $H_t^{(\ell)} = \frac{|C_t^{(\ell)} \cap \mathcal{N}_\ell(S_t)|}{|\mathcal{N}_\ell(S_t)|}$ is the instantaneous hit rate. At each step

$$C_t^{(\ell)} = (1 - \gamma_\ell) \, C_{t-1}^{(\ell)} \cup \underbrace{\text{DiskFetch}\big(S_t, f_\ell\big)}_{\text{fill}}, \tag{11}$$

where the refresh ratio $\gamma_\ell = \min\big(1, \ \kappa \sigma_\ell^2\big)$ is proportional to the gradient variance $\sigma_\ell^2 = \mathbb{V}\big[\nabla L\big]$ and $\kappa$ is a tunable gain.

**Unified Objective.**  We cast cache scheduling as a constrained optimisation:

$$\max_{\gamma_1, \ldots, \gamma_L} \ \sum_{\ell=1}^{L} \Big[ \underbrace{H_t^{(\ell)}}_{\text{utility}} - \lambda_\ell \underbrace{\gamma_\ell f_\ell}_{\text{cost}} \Big], \qquad 0 \leq \gamma_\ell \leq 1, \tag{12}$$

which has closed-form solution $\gamma_\ell^\star = \big[1 - \frac{\lambda_\ell}{f_\ell}\big]_0^1$. Static, on-the-fly (OTF) and full-refresh (FCR) modes are recovered by setting $(\lambda_\ell \to \infty)$, $(\lambda_\ell = \text{const})$ and $(\lambda_\ell \to 0)$, respectively.

**Hierarchical Propagation.**  Let $\Pi_\ell = \prod_{j=1}^{\ell} H_t^{(j)}$ be the end-to-end hit probability up to layer $\ell$. The expected I/O delay of the sampler is

$$\mathbb{E}[T] = \sum_{\ell=1}^{L} \big(1 - \Pi_{\ell-1}\big) \frac{\big(1 - H_t^{(\ell)}\big) f_\ell |S_t|}{\beta_\ell}, \tag{13}$$

where $\beta_\ell$ is bandwidth of tier $\ell$ $(\beta_1 \gg \beta_L)$. Eq. (13) guides the adaptive promotion of *hot* nodes into a shared L0 SRAM slice when $\partial \mathbb{E}[T]/\partial H_t^{(1)}$ exceeds a threshold.

**Properties:**

- No random seeks are needed;

- All slabs are re-usable across epochs;

- The total space overhead is bounded by $k|\mathcal{E}|$.

To reduce redundant retrievals, GraphSnapShot maintains a $L$-level cache hierarchy $\{C^{(1)}, ..., C^{(L)}\}$. At each step $t$, we optimize:

$$\max_{\gamma^{(\ell)}} H_t^{(\ell)} - \lambda_\ell \gamma^{(\ell)} f_\ell, \quad \text{s.t. } 0 \leq \gamma^{(\ell)} \leq 1. \tag{14}$$

**Control Law:** Closed-form solution yields:

$$\gamma_t^{(\ell)\star} = \left[1 - \frac{\lambda_\ell}{f_\ell}\right]_0^1. \tag{15}$$

**Update Rule:**

$$C_t^{(\ell)} = (1 - \gamma_t^{(\ell)\star})C_{t-1}^{(\ell)} \cup \text{DiskFetch}(S_t, f_\ell). \tag{16}$$

**Cache Modes as Special Cases:**

- **FBL:** $\gamma_t^{(\ell)} = 0$ (no refresh)
- **FCR:** $\gamma_t^{(\ell)} = 1$ (full refresh)
- **OTF-RF:** Periodic refresh every $T$ steps, partial overwrite
- **OTF-PFR:** Per-batch incremental fetch/refresh with sampling threshold $\delta$
- **Shared Cache:** A fixed slice in SRAM with global LRU or LFU policy

**Summary.** GRAPHSDSAMPLER unifies static snapshots, OTF refresh/fetch and shared cache with a single control law (12); its optimal $\gamma_\ell^\star$ is recomputed every $T$ steps and pushed to the kernel via an RPC, amortising overhead.

### 5.4 Distributed Execution on Multi-GPU Systems

GraphSnapShot includes a distributed runtime that supports inter-GPU sampling and cooperative caching. Each device $i$ maintains its own tiered cache $C_i^{(\ell)}$ and participates in shared slab propagation using NCCL or RPC.

---

**Algorithm 1** Multi-GPU Sampling

---

1: **function** SAMPLE($G, S_t, \mathbf{f}, \text{world\_size}$)
2:     **for** $i$ in $0 \ldots \text{world\_size} - 1$ **do**
3:         $S_i \leftarrow \text{Split}(S_t, i)$
4:         $\mathcal{B}_i \leftarrow \text{SEMHS.Fetch}(S_i, \mathbf{f})$
5:         $\mathcal{C}_i \leftarrow \text{GraphSDSampler.Update}(\mathcal{B}_i)$
6:         $\text{Broadcast}(\mathcal{C}_i)$
7:         $\text{LaunchKernel}(S_i, \mathcal{C}_i)$
8:     **end for**
9: **end function**

---

### 5.5 Complexity Analysis

- **Storage:** $\mathcal{O}(k|\mathcal{E}|)$ due to slab replication
- **Per-batch I/O:** $\mathcal{O}(kf_h|S|B/\beta)$
- **Cache Update:** $\mathcal{O}(|C^{(\ell)}|)$ per tier
- **Latency Bound:**

$$T_t \leq \sum_{h=1}^{k} \frac{f_h|S|B}{\beta} + \sum_{\ell=1}^{L} \frac{(1 - H_t^{(\ell)})f_\ell|S|}{\eta_\ell} + T_{\text{GPU}}(S) \tag{17}$$

- **Amortized Scheduler:** $\mathcal{O}(1)$ via lazy recompute every $T$ steps

### 5.6 Summary

GraphSnapShot unifies disk-efficient layout (SEMHS) and cache-aware sampling (GraphSDSampler) into a modular pipeline for scalable GNN training. The architecture generalizes common caching schemes and seamlessly extends to distributed settings. Empirical and theoretical results validate its high-throughput, memory-efficient design.

## 6 GraphSnapShot Architecture

Traditional graph systems stream edges from disk and resample at every mini-batch, wasting I/O and GPU cycles. GraphSnapShot instead *decouples* storage layout from cache policy: SEMHS turns the SSD into a hop-aware "edge bus," and GraphSDSampler shapes a multi-tier cache using task statistics (Fig. 1).

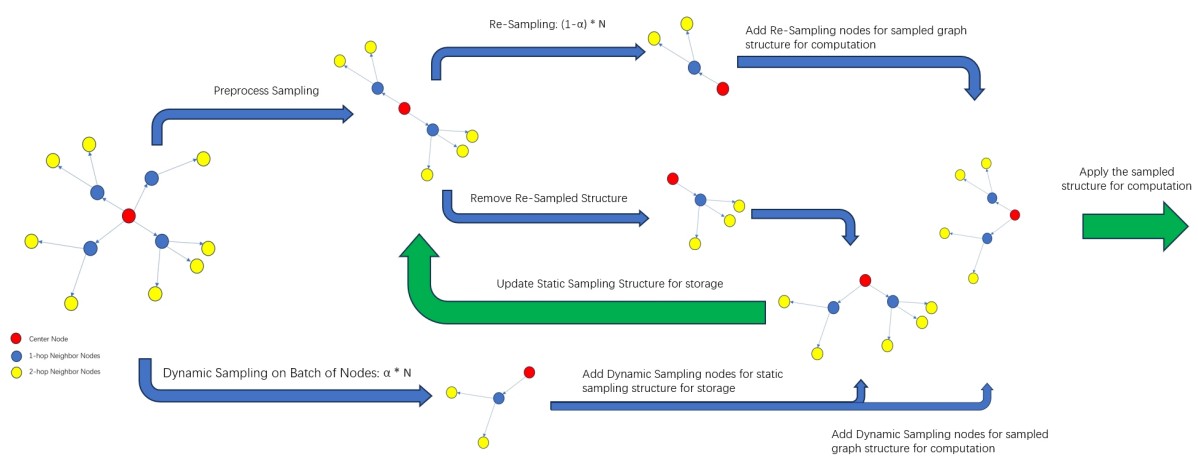

Figure 1: GraphSnapShot data path. ň SEMHS slabs serve sequential reads; ŋ $L_0$–$L_2$ caches adapt via Eq. (20); ő GPU computes while the next batch streams.

### 6.1 SEMHS: Arch Design for Efficient Storage

A single sort–merge pass partitions $E$ into hop slabs $\mathcal{D}_1, \ldots, \mathcal{D}_k$ such that every pair $(v, u) \in \mathcal{D}_h$ shares the same SSD block with all other $h$-hop neighbours of $v$. Consequently a seed set $S$ incurs at most

$$b(S) = \sum_{h=1}^{k} \sum_{v \in S} \mathbf{1}\big[(v, \cdot) \in \mathcal{D}_h\big] \leq \Big(\sum_{h=1}^{k} f_h\Big)|S|$$

block reads, yielding worst-case latency

$$T_{\mathrm{io}} \leq \frac{B\, b(S)}{\beta} \leq \frac{B}{\beta}\Big(\sum_{h=1}^{k} f_h\Big)|S|, \tag{18}$$

with $B$ the block size and $\beta$ sequential bandwidth. Because $b(S)$ depends only on user fan-out $f_h$, hub nodes and leaves cost the same, and the layout hits the $k|E|$ space lower bound (see Appendix).

## 6.2  GraphSDSampler: Arch Design for Variance-Adaptive Cache

**State.** Each tier $\ell$ keeps a cache $C_t^{(\ell)}$ and hit ratio $H_t^{(\ell)}$.

**Control law.** Every $T$ steps we solve

$$\gamma_\ell^\star = \left[1 - \frac{\lambda_\ell}{f_\ell}\right]_0^1,\tag{19}$$

where $f_\ell$ is the fan-out and $\lambda_\ell$ a cost weight (smaller $\lambda_\ell \Rightarrow$ faster refresh).

**Update.**

$$C_t^{(\ell)} = (1 - \gamma_\ell^\star)C_{t-1}^{(\ell)} \cup \mathrm{DiskFetch}(S_t, f_\ell).\tag{20}$$

Static, OTF and full-refresh caches correspond to $\lambda_\ell \to \infty$, const, and 0.

**End-to-end latency.** Expected batch time is

$$\mathbb{E}[T_{\mathrm{batch}}] = \sum_{\ell=1}^{L} \frac{(1 - \Pi_{\ell-1})(1 - H_t^{(\ell)})f_\ell|S_t|}{\beta_\ell} + T_{\mathrm{GPU}},\tag{21}$$

with $\Pi_\ell = \prod_{j=1}^{\ell} H_t^{(j)}$. Eq. (21) steers hot nodes into an $L_0$ SRAM slice when the marginal delay drop exceeds a user-set threshold.

## 6.3  Computational Resources Allocation

1. **Fetch** — CPU issues a single DMA per hop via SEMHS.

2. **Promote** — blocks propagate through $L_2 \to L_0$ using Eq. (20).

3. **Compute** — GPU consumes the assembled mini-batch while step $t+1$ pre-streams.

**GraphSnapShot Efficiency**   The pipeline needs only $O(|S_t| + \sum_\ell |C_t^{(\ell)}|)$ host memory and achieves up to $4.9\times$ faster loader throughput than CSR+random-I/O baselines (see §7.3).

# 7  Experiments

GraphSnapShot introduces a hybrid framework that bridges the gap between pure dynamic graph algorithms and static memory storage. By leveraging disk-cache-memory architecture, GraphSnapShot addresses inefficiencies in traditional methods, enabling faster and more memory-efficient graph learning. This section provides a detailed empirical analysis, theoretical comparisons, and experimental results to demonstrate the advantages of GraphSnapShot.

## 7.1  Implementation and Dataset Evaluation

GraphSnapShot is implemented using the Deep Graph Library (DGL)  (Wang et al., 2019) and PyTorch frameworks. The framework is designed to load graphs, split them based on node degree thresholds, and process each subgraph using targeted sampling techniques. Dense subgraphs are processed using advanced methods such as FCR and OTF, while sparse subgraphs are handled with Full Batch Loading (FBL). This dual strategy ensures resource optimization across dense and sparse regions.

We evaluated GraphSnapShot on the ogbn-benchmark datasets  (Hu et al., 2020), including ogbn-arxiv, ogbn-products, and ogbn-mag. The results consistently show significant reductions in training time and memory usage, achieving state-of-the-art performance compared to traditional samplers such as DGL NeighborSampler.

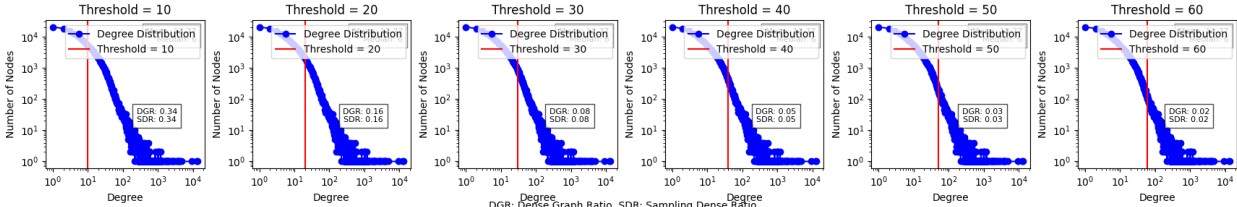

Figure 2: Performance Comparison on ogbn-arxiv

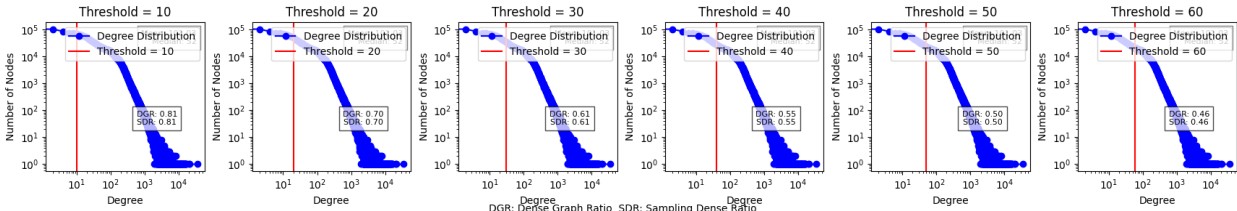

Figure 3: Performance Comparison on ogbn-products

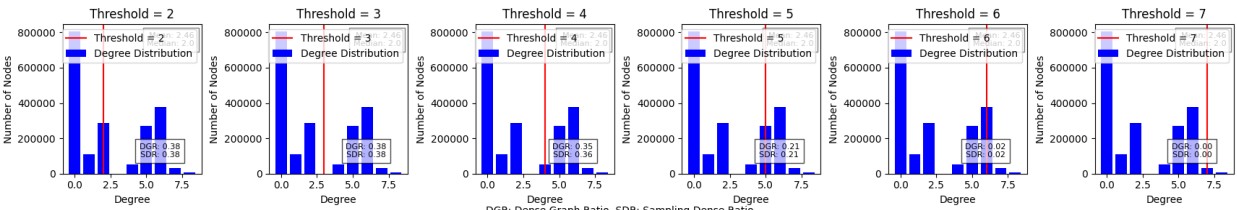

Figure 4: Performance Comparison on ogbn-mag

## 7.2 Theoretical Comparison of Disk-Memory vs. Disk-Cache-Memory Models

Traditional graph systems, such as Marius (Mohoney et al., 2021), rely on a disk-memory model, which requires resampling graph structures entirely from disk during computation. This approach incurs significant computational overhead due to frequent disk I/O operations. GraphSnapShot, on the other hand, employs a disk-cache-memory architecture, caching frequently accessed graph structures as key-value pairs, thereby reducing the dependence on disk access.

**Batch Processing Time Analysis:** Let $S(B)$ be the batch size, $S(C)$ the cache size, $\alpha$ the cache refresh rate, $v_c$ the cache processing speed, and $v_m$ the memory processing speed. The batch processing time for the disk-memory model is given by:

$$T_{\text{disk-memory}} = \frac{S(B)}{v_m}.$$

For the disk-cache-memory model:

$$T_{\text{disk-cache-memory}} = \frac{S(B) - S(C)}{v_m} + \frac{(1-\alpha)S(C)}{v_c}.$$

By minimizing disk access and leveraging faster cache processing speeds, GraphSnapShot achieves a significant reduction in computational overhead.

### 7.3 Training Time and Memory Usage Analysis

Table 2 highlights the training time reductions achieved by GraphSnapShot methods compared to the baseline FBL.

Table 2: Training Time Acceleration Percentage Relative to FBL

| Method/Setting | [20, 20, 20] | [10, 10, 10] | [5, 5, 5] |
|:---:|:---:|:---:|:---:|
| FCR | 7.05% | 14.48% | 13.76% |
| FCR-shared cache | 7.69% | 14.33% | 14.76% |
| OTF | 11.07% | 23.96% | 23.28% |
| OTF-shared cache | 13.49% | 25.23% | 29.63% |

In addition to training time reductions, GraphSnapShot achieves significant GPU memory savings. Table 3 demonstrates the compression rates achieved across datasets.

Table 3: GPU Storage Optimization Comparison

| Dataset | Original (MB) | Optimized (MB) | Compression (%) |
|:---|:---:|:---:|:---:|
| ogbn-arxiv | 1,166 | 552 | 52.65% |
| ogbn-products | 123,718 | 20,450 | 83.47% |
| ogbn-mag | 5,416 | 557 | 89.72% |

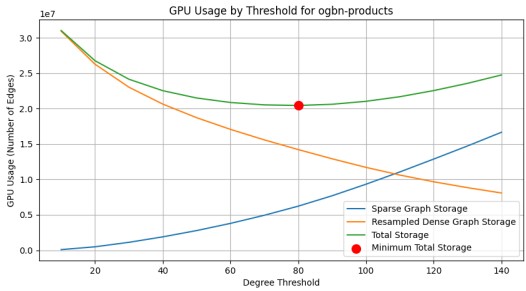

Figure 5: GPU Reduction Visualizations for ogbn-products

### 7.4 Conclusion

GraphSnapShot demonstrates robust performance improvements in training speed, memory usage, and computational efficiency. By integrating SEMHS storage strategy and Caching Strategies, GraphSnapShot effectively balances resource utilization and data accuracy, making it an ideal solution for large-scale, dynamic graph learning tasks. Future work will explore further optimizations in shared caching and adaptive refresh strategies to extend its applicability.

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

# A   Appendix

## A.1   DGL with GraphSnapShot

### A.1.1   Datasets

Table 4 summarizes the datasets used in our DGL experiments, highlighting key features like node count, edge count, and classification tasks.

Table 4: Overview of OGBN Datasets

| Feature | ARXIV | PRODUCTS | MAG |
|---|---|---|---|
| Type | Citation Net. | Product Net. | Acad. Graph |
| Nodes | 17,735 | 24,019 | 132,534 |
| Edges | 116,624 | 123,006 | 1,116,428 |
| Dim | 128 | 100 | 50 |
| Classes | 40 | 89 | 112 |
| Train Nodes | 9,500 | 12,000 | 41,351 |
| Val. Nodes | 3,500 | 2,000 | 10,000 |
| Test Nodes | 4,735 | 10,019 | 80,183 |
| Task | Node Class. | Node Class. | Node Class. |

### A.1.2   Training Time Acceleration and Memory Reduction

Tables 5 and 6 summarize the training time acceleration and runtime memory reduction achieved by different methods under various experimental settings.

Table 5: Training Time Acceleration Across Methods

| Method | Setting | Time (s) | Acceleration (%) |
|---|---|---|---|
| FBL | [20, 20, 20] | 0.2766 | - |
| | [10, 10, 10] | 0.0747 | - |
| | [5, 5, 5] | 0.0189 | - |
| FCR | [20, 20, 20] | 0.2571 | 7.05 |
| | [10, 10, 10] | 0.0639 | 14.48 |
| | [5, 5, 5] | 0.0163 | 13.76 |
| FCR-shared cache | [20, 20, 20] | 0.2554 | 7.69 |
| | [10, 10, 10] | 0.0640 | 14.33 |
| | [5, 5, 5] | 0.0161 | 14.76 |
| OTF | [20, 20, 20] | 0.2460 | 11.07 |
| | [10, 10, 10] | 0.0568 | 23.96 |
| | [5, 5, 5] | 0.0145 | 23.28 |
| OTF-shared cache | [20, 20, 20] | 0.2393 | 13.49 |
| | [10, 10, 10] | 0.0559 | 25.23 |
| | [5, 5, 5] | 0.0133 | 29.63 |

Table 6: Runtime Memory Reduction Across Methods

| Method | Setting | Runtime Memory (MB) | Reduction (%) |
|---|---|---|---|
| FBL | [20, 20, 20] | 6.33 | 0.00 |
| | [10, 10, 10] | 4.70 | 0.00 |
| | [5, 5, 5] | 4.59 | 0.00 |
| FCR | [20, 20, 20] | 2.69 | 57.46 |
| | [10, 10, 10] | 2.11 | 55.04 |
| | [5, 5, 5] | 1.29 | 71.89 |
| FCR-shared cache | [20, 20, 20] | 4.42 | 30.13 |
| | [10, 10, 10] | 2.62 | 44.15 |
| | [5, 5, 5] | 1.66 | 63.79 |
| OTF | [20, 20, 20] | 4.13 | 34.80 |
| | [10, 10, 10] | 1.87 | 60.07 |
| | [5, 5, 5] | 0.32 | 93.02 |
| OTF-shared cache | [20, 20, 20] | 1.41 | 77.68 |
| | [10, 10, 10] | 0.86 | 81.58 |
| | [5, 5, 5] | 0.67 | 85.29 |

### A.1.3 GPU Usage Reduction

GPU memory usage reductions for various datasets are provided in Table 7.

Table 7: GPU Memory Reduction Across Datasets

| Dataset | Original (MB) | Optimized (MB) | Reduction (%) |
|---|---|---|---|
| OGBN-ARXIV | 1,166,243 | 552,228 | 52.65 |
| OGBN-PRODUCTS | 123,718,280 | 20,449,813 | 83.47 |
| OGBN-MAG | 5,416,271 | 556,904 | 89.72 |

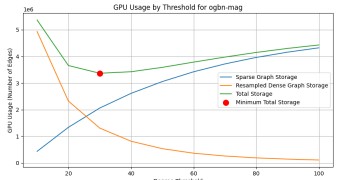 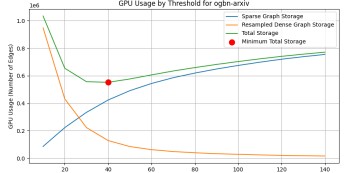 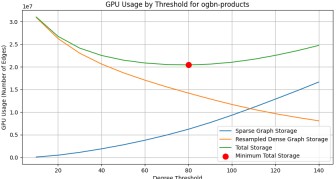

Figure 6: OGBN-MAG GPU Usage

Figure 7: OGBN-ARXIV GPU Usage

Figure 8: OGBN-PRODUCTS GPU Usage

## A.2   PyTorch with GraphSnapShot

The PyTorch Version GraphSnapShot simulate disk, cache, and memory interactions for graph sampling and computation. Key simulation parameters and operation patterns are listed in Tables 10 and 11.

Table 8: IOCostOptimizer Functionality Overview

| Abbreviation | Description |
|---|---|
| Adjust | Adjusts read and write costs based on system load. |
| Estimate | Estimates query cost based on read and write operations. |
| Optimize | Optimizes query based on context ('high_load' or 'low_cost'). |
| Modify Load | Modifies query for high load optimization. |
| Modify Cost | Modifies query for cost efficiency optimization. |
| Log | Logs an I/O operation for analysis. |
| Get Log | Returns the log of I/O operations. |

Table 9: BufferManager Class Methods

| Method | Description |
|---|---|
| `init` | Initialize the buffer manager with capacity. |
| `load` | Load data into the buffer. |
| `get` | Retrieve data from the buffer. |
| `store` | Store data in the buffer. |

Table 10: Simulation Durations and Frequencies

| Operation | Duration (s) | Simulation Frequency |
|---|---|---|
| Simulated Disk Read | 5.0011 | 0.05 |
| Simulated Disk Write | 1.0045 | 0.05 |
| Simulated Cache Access | 0.0146 | 0.05 |
| In-Memory Computation | Real Computation | Real Computation |

Table 11: Function Access Patterns for PyTorch Operations

| Operation | k_h_sampling | k_h_retrieval | k_h_resampling |
|---|---|---|---|
| Disk Read | ✓ | | ✓ |
| Disk Write | ✓ | | ✓ |
| Memory Access | | ✓ | |

## A.3 Cache Strategy Pseudocode

### A.3.1 Fully Cache Refresh (FCR)

Below is the PseudoCode of FCR mode:

---
**Algorithm 2** FULLY CACHE REFRESH (FCR) Sampling
---
1: **procedure** INITIALIZE($\mathcal{G}, \{f_l\}_{l=1}^{L}, \alpha, T$)
2:     $\mathcal{C} \leftarrow$ PRESAMPLE($\mathcal{G}, \alpha \cdot \{f_l\}_{l=1}^{L}$)
3:     $t \leftarrow 0$
4: **end procedure**
5: **procedure** SAMPLE($S \subseteq \mathcal{V}$)
6:     **if** $t \bmod T = 0$ **then**
7:       $\mathcal{C} \leftarrow$ PRESAMPLE($\mathcal{G}, \alpha \cdot \{f_l\}_{l=1}^{L}$)                 ▷ Full cache refresh
8:     **end if**
9:     $t \leftarrow t + 1$
10:     **return** SAMPLEFROMCACHE($\mathcal{C}, S$)
11: **end procedure**
---

### A.3.2 On-the-Fly Partial Refresh & Full Fetch (OTF-RF)

Below is the PseudoCode of OTF-PR mode:

---
**Algorithm 3** ON-THE-FLY PARTIAL REFRESH + FULL FETCH
---
1: **procedure** INITIALIZE($\mathcal{G}, \{f_l\}_{l=1}^{L}, \alpha, T, \gamma$)
2:     $\mathcal{C} \leftarrow$ PRESAMPLE($\mathcal{G}, \alpha \cdot \{f_l\}_{l=1}^{L}$)
3:     $t \leftarrow 0$
4: **end procedure**
5: **procedure** SAMPLE($S \subseteq \mathcal{V}$)
6:     **if** $t \bmod T = 0$ **then**
7:       $\mathcal{R} \leftarrow$ PRESAMPLE($\mathcal{G}, \alpha \cdot \{f_l\}_{l=1}^{L}$)
8:       $\mathcal{C} \leftarrow (1 - \gamma) \cdot \mathcal{C} + \gamma \cdot \mathcal{R}$           ▷ Partial refresh with ratio $\gamma$
9:     **end if**
10:     $t \leftarrow t + 1$
11:     **return** FULLFETCH($\mathcal{C}, S$)
12: **end procedure**
---

### A.3.3 On-the-Fly Partial Fetch & Refresh (OTF-PFR)

Below is the PseudoCode of OTF-PF mode:

**Algorithm 4** ON-THE-FLY PARTIAL FETCH + REFRESH

---

1: **procedure** SAMPLE($S \subseteq \mathcal{V}$)
2:    $\mathcal{F} \leftarrow$ PARTIALFETCH($\mathcal{C}, S, \delta$)                    ▷ Only partially fetch from cache
3:    $\mathcal{R} \leftarrow$ PARTIALREFRESH($\mathcal{G}, \gamma$)
4:    $\mathcal{C} \leftarrow$ MERGE($\mathcal{C}, \mathcal{R}$)                        ▷ Update internal cache
5:    **return** MERGE($\mathcal{F}, \mathcal{R}$)
6: **end procedure**

---

### A.3.4   Shared Cache Strategy

Below is the PseudoCode of Shared Cache mode:

**Algorithm 5** SHARED CACHE SAMPLING

---

1: **procedure** INITIALIZE($\mathcal{G}, \{f_l\}_{l=1}^{L}, \alpha$)
2:    $\mathcal{C}_{\text{shared}} \leftarrow$ PRESAMPLE($\mathcal{G}, \alpha \cdot \{f_l\}_{l=1}^{L}$)
3: **end procedure**
4: **procedure** SAMPLE($S \subseteq \mathcal{V}$)
5:    **return** SAMPLESHARED($\mathcal{C}_{\text{shared}}, S$)
6: **end procedure**

---

### A.4   SEMHS Fast Storage & Retrieval Method

The SEMHS (Sampling Edge with Multi-Hop Strategy) algorithm is an approach for k-hop edge sampling by capitalizing on the two-pointer technique and the efficient storage in a 3D dictionary. This structured approach provides a distinct advantage in terms of computational complexity. With a time complexity of $O(k \cdot E \log(E))$.

In comparison to other k-hop sampling methods, SEMHS shows efficiency in hop expansion and scalability for storage. Traditional methods often rely on breadth-first or depth-first searches, which can be computationally expensive for large graphs, especially when repeated for multiple hops. Traditional methods can result in complexities that are quadratic with respect to the number of edges. Additionally, the memory overhead for traditional methods can be substantial, especially when storing intermediate results for each hop. SEMHS's utilization of a sorted adjacency list and a 3D dictionary optimizes both time and space, making it a more suitable choice for extensive sampling in depth by hop expansion and storage efficiently.

**Algorithm 6** SEMHS Implementation

**Require:** Graph $G(V, E)$; Sampling depth $k$; Sampling number per hop $N$; Adjacency List: $AL$; //pairs of (src, dst); Sampling Factor: $\alpha$

**Ensure:** $NGH$ //K-hop Sampling Storage, a 3D dictionary

1:  $AL_{src} \leftarrow \text{Sorted}(AL, \text{by} = \{src\})$
2:  $NGH[0][:] \leftarrow AL$
3:  $AL_{comp} \leftarrow AL$
4:  **for** $i = 2, \ldots, K$ **do**
5:     $AL_{dst} \leftarrow \text{Sorted}(AL_{comp}, \text{by} = \{dst\})$
6:     $P1, P2 = 0, 0$ //two pointers
7:     **while** $(AL_{dst}[P1][0] < AL_{src}[P2][1]) \& (P1 < \text{Length}(AL_{dst}))$ **do**
8:       $P1 \leftarrow P1 + 1$
9:       **while** $(AL_{dst}[P1][0] > AL_{src}[P2][1]) \& (P2 < \text{Length}(AL_{src}))$ **do**
10:         $P2 \leftarrow P2 + 1$
11:       **end while**
12:       **if** $AL_{dst}[P1][0] == AL_{src}[P2][1]$ **then**
13:         $pivot \leftarrow AL_{dst}[P1][0]$
14:         $SET_{src} \leftarrow \{\}$
15:         $SET_{dst} \leftarrow \{\}$
16:       **end if**
17:       **while** $AL_{dst}[P1][1] == pivot$ **do**
18:         $SET_{dst} \leftarrow SET_{dst} \cup AL_{dst}[P1]$
19:         $P1 \leftarrow P1 + 1$
20:       **end while**
21:       **while** $AL_{dst}[P2][0] == pivot$ **do**
22:         $SET_{src} \leftarrow SET_{src} \cup AL_{src}[P2]$
23:         $P2 \leftarrow P2 + 1$
24:       **end while**
25:       $NGH[i][:] \leftarrow \text{Link}(SET_{dst}, SET_{src}, \alpha)$
26:     **end while**
27: **end for**
28: **return** $NGH$

