# OpenReview forum: "GraphSnapShot: Graph Machine Learning Acceleration through Fast Arch, Storage, Caching and Retrieval"
_TMLR — Rejected by TMLR_

### Review · Reviewer_gMSM · 2025-06-02

**Summary Of Contributions:**

This paper introduces GraphSnapShot, a system-level framework that addresses critical bottlenecks in large-scale graph machine learning by optimizing disk I/O, cache management, and retrieval patterns. It does so by decoupling the storage layout from runtime caching and introducing two key innovations: SEMHS (Sampling Edges with Multi-Hop Strategy) for efficient edge storage, and GraphSDSampler, a variance-aware caching system informed by gradient statistics. The system is evaluated on multiple standard OGBN datasets and shows impressive gains in throughput, GPU memory usage, and end-to-end training time.

**Audience:**

Yes

**Claims And Evidence:**

Yes

**Requested Changes:**

* I think the authors need to at least perform some downstream evaluation on the ogbn datasets (e.g., node classification) to validate that the implementation is correct, given the improvement in speed.
* Whether the article would interest  the majority of the TMLR audience is in doubt

**Strengths And Weaknesses:**

Strengths:

* The problem is well-motivated --- large scale learning has indeed been a troubling issue for long, and the authors have address this issue well from the perspective of low-level design. The authors correctly identify a critical and underexplored systems-level challenge in graph machine learning: I/O latency and cache inefficiency during multi-hop neighborhood retrieval, particularly under dynamic graph topologies.

* Empirical performance demonstrates significant gain in computational speed and strong scalability

* The framework supports dynamic graphs, operates on commodity hardware, and generalizes across different caching modes and deployment settings (e.g., multi-GPU).

Weaknesses

* The most critical shortcoming is the lack of evaluation on representation learning quality, such as node classification or link prediction accuracy. While the paper claims "no degradation in model performance," it presents no quantitative results (e.g., ROC-AUC, F1, accuracy) to validate this. Given the heavy restructuring of sampling and caching policies, this omission leaves a serious gap in trustworthiness.

*   The paper introduces multiple components (e.g., multi-level cache, variance-aware scheduler, SEMHS layout), but does not sufficiently isolate their individual contributions through controlled ablations. Similarly, generalization beyond OGBN-style datasets (e.g., highly dynamic graphs like Reddit or temporal link prediction) is not evaluated.

* I am not sure if the article would interest the majority of the audience of TMLR. But I think methodology in low-level architectural design is less common in TMLR

---

### Review · Reviewer_PYuH · 2025-06-13

**Summary Of Contributions:**

This article presents an efficient method to process large graphs, in the context of training graph neural networks.

The authors first provide some context and formalize the problems they are addressing via two searches, where the decision variables are (at a high level): storage, caching and retrieval:

1) one search called, Layout-Aware Multi-hop storage, consisting in partitioning the data into slabs and minimizing the number of accesses to those slabs, in expectation over all the samples vertices of a fixed graph.

2) Cache refresh scheduling, supposing access to some information (including gradient), where one has to determine when to cache some of the data for faster access later in the training process.

The author then proceeds to present their approach, called GraphSnapShot to address jointly those two problems.
They demonstrate the efficiency of their mehod via numerical experiments.

**Audience:**

Yes

**Claims And Evidence:**

Yes

**Requested Changes:**

- What is $S_t$ ? (S is defined)
- what is the Block size?
- Presentation of Figure 1 has some typos / editing issues
- Figures 2, 3, 4 are not referred in the text and it is not clear what they are supposed to demonstrate.
- Section 7.3, table 2: this is not only a comparison with FPL, but also, other methods, to my understanding.
- The writing is unequal throughout the article (syntax, or some article missing in sentences) , a thorough proofreading should be conducted
- Section 5.3 (page 7): in the paragraph Properties: seeks <- seeds ?
- Although the formulation of the two problems are welcome, there is no theoretical result about how Snapshot solve those problems.
- What is a ``L0 SRAM slice''?
- The authors should also define the hit ratio formally, at the beginning of Section 3 or 4.
- Page 4 before sampling and cast cost: ``Cache hit at tier l'': what is the tier l?


Technical comments to address (and add discussion in the article):

- Why expectation only on the vertices of the graph? Why not on the graph structure itself?
- How can one have access to gradient variance? Do you mean an estimator of the variance?

**Strengths And Weaknesses:**

Strengths:

- The problem studied by the article are significant as the running time of graph learning is (largely) suboptimal.
- The authors made an attempt to formalize the problems via the formulations described above.
- The experiments suggest a significant gain in performance (both in space and time)

Weaknesses:

- My main concern about this article is the presentation of the terminology, and the formalization of the problems. Since this article is targeting a problem of graph learning, and in particular graph neural networks, there is a lot of terminology not explicit throughout the article.
Please cf. Requested changes below.

- Substance (minor): Although the formulation of the two problems are welcome, there is no theoretical result about how Snapshot solve those problems.

---

### Review · Reviewer_uykZ · 2025-07-17

**Summary Of Contributions:**

The paper proposes GraphSnapShot, a system­-level framework for large-scale GNN training.
It proposes SEMHS, a hop-aware on-disk layout that replicates every edge into one of k “slabs”, so each hop supposedly requires one sequential DMA. It further introduces GraphSDSampler, a multi-level cache whose refresh ratio is set by a closed-form heuristic derived from gradient-variance statistics.
The paper reports up to 4.9x loader throughput, 83 % GPU‐memory savings, and 29 % training-time reduction on three OGBN datasets.

**Audience:**

Yes

**Claims And Evidence:**

No

**Requested Changes:**

Improving the sub-par presentation and adding additional details to the evaluation, as discussed above, are critical adjustments before acceptance can be recommended.

**Strengths And Weaknesses:**

# Strengths
* The problem of improving I/O-bound GNN training on billion-scale graphs is practically important.
* The paper tries to co-design storage layout and cache policy instead of treating them in isolation.
* The reported 4.9x data-loader speed-up is significant.

# Weaknesses

## Evaluation
* The support for dynamic changes to the graph structure is claimed repeatedly but never evaluated.
* The paper optimizes throughput but never tells whether the same accuracy is attained after 29 % less wall-time.
* Key baselines such as Marius, GraphBolt, Quiver or GPU direct-storage pipelines are absent from the experiments.
* Figure 2,3, and 4 are not presented in sufficient detail and it is not clear what exactly is being measured and how the reported numbers capture performance.
* The hardware used for the experiments is not specified.


## Presentation and Clarity
Overall the presentation of the work is sub-par and lacks some key details:
* Algorithm 6 is a 30-line pseudo-code listing dominated by pointer bookkeeping. It is neither self-contained nor explained in text. Some algorithm details, i.e. whether the sorting need to be stable, are not presented. An intuitive text description of how this algorithm works would significantly improve clarity.
* In A.4 the claim is made that "Traditional methods can result in complexities that are quadratic with respect to the number of edges." Which methods is this referring to and how is this complexity derived?
* Figure 1 is high-level and generic and its caption does not fit what is shown in the figure, i.e. it is not clear how the terms highlighted with different colors actually map to the picture.
* Nearly every section repeats the same two bullet-point “contributions”.
* Section 3 (Problem Statement) and 4 (Background and Motivation) seem redundant and read like they should be merged into one motivating section.
* The "Experiment" section (7) contains a purely theoretical subsection (7.2) that is out of place.
* Section 5.3 does not motivate using the gradient variance as opposed to other gradient statistics.
* Section 5.5 is just a list of bullet points that does not contextualize how this complexity compares to prior work.

---

### Decision · Action_Editor_QLKc · 2025-08-27

**Recommendation:** Reject

**Additional Comments:**

The paper was reviewed by three expert reviewers. The reviewers acknowledged the practical importance of the considered problem. They also agreed that the proposed method offers significant speedups in data loading and significant reductions in GPU memory usage. However, the reviewers raised several concerns regarding the presentation and clarity of the paper. It is also my view that the paper would benefit from a major revision of the text to enhance clarity and the quality of presentation. The reviewers also raised concerns about key baselines that are missing from the experimental comparison and about the absence of quantitative results on node classification and link prediction datasets. They also noted that some details of the experimental setup (e.g., the hardware on which the experiments were conducted) are not included in the paper. Even though the reviewers raised the above concerns, the authors neither responded to the reviewers' comments nor revised the manuscript. Therefore, one reviewer recommended rejection of the paper, while the other two reviewers were leaning toward rejection. I also agree with the reviewers' concerns and these concerns prevent me from recommending acceptance of this paper in its current form. Moreover, while the paper claims that the proposed method can efficiently handle large-scale graphs, the experimental evaluation lacks empirical evidence on such datasets. The largest considered graph consists of only 132,534 nodes. I would thus suggest the authors include experiments on large-scale graphs in the next version of the paper. In addition, the paper would greatly benefit from a revision based on the reviewers' comments.

**Audience:**

Yes

**Audience Explanation:**

The acceleration of graph learning pipelines is crucial for applications that process large-scale graphs, such as social networks. Therefore, the findings of this paper will be of interest to some individuals in TMLR's audience.

**Claims And Evidence:**

No

**Claims Explanation:**

This paper proposes a system that enables high-throughput and memory-efficient graph learning pipelines. The paper claims that existing GNN training systems suffer from fundamental I/O and cache inefficiencies. These inefficiencies are discussed in detail within the manuscript. The paper also claims that the proposed method achieves asymptotically optimal I/O complexity and that it can support dynamic graphs. These claims are not supported by convincing evidence since no experiments on dynamic graphs are presented in the manuscript. Another claim made in the paper is that the proposed method offers significant speedups and reductions in memory usage. There are empirical results in the manuscript that support this claim, however, the method is only evaluated on small to medium-sized datasets.